# Late Dialysis Modality Education Could Negatively Predict Peritoneal Dialysis Selection

**DOI:** 10.3390/jcm11144042

**Published:** 2022-07-13

**Authors:** Takashin Nakayama, Ken Nishioka, Kiyotaka Uchiyama, Kohkichi Morimoto, Ei Kusahana, Naoki Washida, Shintaro Yamaguchi, Tatsuhiko Azegami, Tadashi Yoshida, Hiroshi Itoh

**Affiliations:** 1Division of Endocrinology, Metabolism and Nephrology Department of Internal Medicine, Keio University School of Medicine, 35 Shinanomachi, Shinjuku-ku, Tokyo 160-8582, Japan; takashin.nakayama@gmail.com (T.N.); nishiken-ken@hotmail.co.jp (K.N.); shoot.8.9387@gmail.com (E.K.); washida@iuhw.ac.jp (N.W.); yama1005@a6.keio.jp (S.Y.); t.azegami-1114@z2.keio.jp (T.A.); hiito@keio.jp (H.I.); 2Apheresis and Dialysis Center, Keio University School of Medicine, 35 Shinanomachi, Shinjuku-ku, Tokyo 160-8582, Japan; kohkichi.morimoto@keio.jp (K.M.); tayoshida-npr@umin.ac.jp (T.Y.); 3Department of Nephrology, International University of Health and Welfare Narita Hospital, 4-3 Kozunomori, Narita, Chiba 286-8686, Japan; 4Keio University Health Center, 4-1-1 Hiyoshi, Kohoku-ku, Yokohama 223-8521, Kanagawa, Japan

**Keywords:** peritoneal dialysis, renal replacement therapy, patient education

## Abstract

Patients with end-stage renal disease are less likely to choose peritoneal dialysis (PD) as renal replacement therapy (RRT). The reasons for this biased selection are still poorly understood. In this study, we evaluated the effect of the timing of RRT education on PD selection. This single-center retrospective observational study included patients who initiated maintenance dialysis at our hospital between April 2014 and July 2021. A logistic regression analysis was performed to investigate the association of RRT education timing with PD selection. Among the 355 participants (median age [IQR] 70 (59–79) years; 28.7% female), 53 patients (14.9%) and 302 patients (85.1%) selected PD and hemodialysis, respectively. Multivariate analysis demonstrated that high estimated glomerular filtration (eGFR) at RRT education positively predicted PD selection (*p* < 0.05), whereas old age (*p* < 0.01) and high Charlson comorbidity index (*p* < 0.05) were negative predictors of PD selection. Female sex (*p* = 0.44), welfare public assistance (*p* = 0.78), living alone (*p* = 0.25), high geriatric nutritional risk index (*p* = 0.10) and high eGFR at first visit to the nephrology department (*p* = 0.83) were not significantly associated with PD selection. Late RRT education could increase the biased selection of dialysis modality.

## 1. Introduction

In recent years, the number of patients with end-stage renal disease (ESRD) worldwide has risen dramatically, and the disease is becoming a threat to public health [1]. Patients with ESRD require renal replacement therapy (RRT), choosing between dialysis or kidney transplantation to survive. In comparison to hemodialysis (HD), peritoneal dialysis (PD) preserves residual renal function, maintains hemodynamic stability, and provides a high quality of life, while being cost-effective [2,3,4,5,6,7,8]. However, despite these advantages, the prevalence rate for PD remains low [9]. In Japan, only 3% patients choose this treatment option, although there is no difference in the medical expenses borne by patients between dialysis modalities [10].

Young age, few concomitant diseases, good nutritional status, full-time employment, and high education level have been reported to be linked to a higher probability of PD selection in patients with ESRD [11,12,13]. Although the link between medical engagement and PD selection is yet unknown, previous research has shown that RRT education remarkably increases the rate of PD selection [14]. However,, few studies have evaluated the effect of the timing of RRT education on dialysis modality selection. The 2015 Kidney Disease Outcomes Quality Initiative guidelines recommend that patients who reach chronic kidney disease (CKD) stage 4 (glomerular filtration rate [GFR], 30 mL/min/1.73 m^2^) should receive RRT education [15]. However, the effect of such early patient education on dialysis modality selection is not fully validated.

To investigate the influence of the timing of RRT education on PD selection, we present a retrospective cohort study comprising patients starting maintenance dialysis at our institution.

## 2. Materials and Methods

### 2.1. Study Population

This single-center, retrospective study, and all its protocols, were reviewed and approved by the Keio University School of Medicine Ethics Committee (Approval No. 20211137). Informed consent was obtained in the form of opt-out on the website. Patients who initiated maintenance dialysis as RRT at Keio University Hospital between April 2014 and July 2021 were enrolled in this study. Patients were excluded if dialysis was started within six months of their first visit to the nephrology department as they could not afford the time to receive proper RRT education. Patients who had undergone a kidney transplant were also excluded.

### 2.2. Data Collection and Participant Evaluation

Data on age, sex, receiving welfare public assistance or not (household income is below the minimum cost of living), living alone or with roommates, smoking history, height, weight, body mass index (BMI), blood pressure, primary disease of ESRD (diabetic kidney disease, glomerulonephritis, nephrosclerosis, polycystic kidney disease, tubulointerstitial nephritis, and others), and comorbidities (diabetes mellitus, hypertension, cerebrovascular disease, coronary artery disease, chronic heart failure and malignancy) were collected. As previously reported, the Charlson comorbidity index (CCI) was calculated from the records [16]. Moreover, we gathered information on serum levels of creatinine, urea nitrogen, albumin, potassium, calcium, phosphorus, triglyceride, high–density lipoprotein (HDL)–cholesterol and low–density lipoprotein (LDL)–cholesterol, and hemoglobin level at dialysis initiation. The calcium level was corrected for the lower range of albumin using Payne’s formula [17]. Geriatric nutritional risk index (GNRI) was calculated from patients’ body weight and serum albumin levels [18]. Estimated glomerular filtration rate (eGFR) was also calculated using the following equation for Japanese people: eGFR = 194 × Creatine^−1.094^ × Age^−0.287^ (×0.739 if female) [19].

We also collected eGFR at the first visit to the nephrology department, and patient education from the electronic medical records. All patients (and their family if appropriate) received RRT education by either, or both, nephrologists and/or nurse specialists before the initiation of maintenance dialysis. The timing of RRT education, and who would primarily provide it, were determined by attending nephrologists. The different forms of dialysis therapy and their detailed features, including the effect on daily life or possible complications that may arise, were discussed in RRT education. Furthermore, data on the timing of general education on CKD by nurse specialists (except RRT education) were also collected. Nurse specialists provided the following general knowledge about CKD in accordance with attending nephrologists’ instructions: dietary habits, medication adherence, living environment modifications and future uremia symptoms. Moreover, the PD selection rate and the timing of education were investigated for each attending doctor.

### 2.3. Statistical Analyses

Continuous variables were expressed as means ± standard deviation or median (25th–75th percentile) according to normality tested by the Shapiro–Wilk test, and binary variables were expressed as percentages. The unpaired Student’s t-test (continuous variables with normal distribution), Mann–Whitney U test (continuous variables without normal distribution), and Chi-squared test (binary variables) were used to compare groups with PD and HD.

Logistic regression analysis was performed to evaluate the odds ratios (OR) and 95% confidence intervals for PD selection as dialysis modality. Because the aim of this study was to clarify the effect of the timing of RRT education on PD selection, eGFR at RRT education was entered into the regression model. Moreover, variables that have been previously reported to be associated with PD selection were included. Age, sex, welfare public assistance, living alone, CCI, GNRI, and eGFR at first visit to the nephrology department were included as independent variables (model 1). We also assumed that decline rate in renal function or renal function at dialysis initiation could be associated with dialysis modality selection. Therefore, additional multivariate models using variables in model 1 + eGFR decline rate for 6 months before dialysis initiation (model 2), and variables in model 2 + eGFR at dialysis initiation (model 3), were created.

All analyses were performed using IBM SPSS Statistics version 27 (IBM, Armonk, NY, USA). All *p*-values were two-sided and *p*-values < 0.05 were considered statistically significant.

## 3. Results

### 3.1. Patient Characteristics

A total of 462 patients initiated maintenance dialysis during the study period (Figure 1). Nine patients had previously undergone kidney transplantation, and 98 patients started dialysis within 6 months of their first visit to the nephrology department. Therefore, we excluded 107 patients based on these criteria. In the end, 355 patients were included in this study. Among them, 53 patients (14.9%) and 302 patients (85.1%) started PD and HD as RRT, respectively. Baseline characteristics of the patients are shown in Table 1. The median age of patients was 70 (59–79) years, and 28.7% were female. 3.7% of patients received welfare public assistance, and 19.4% lived alone. The most common underlying disease among the patients was diabetic kidney disease (36.6%), followed by nephrosclerosis (24.8%), glomerulonephritis (16.6%), tubulointerstitial nephritis (4.2%), and polycystic kidney disease (3.9%). Patients with PD were significantly younger (*p* < 0.01), had higher BMI (*p* < 0.05) and lower CCI (*p* < 0.01), and were more likely to have glomerulonephritis as the underlying disease (*p* < 0.05), compared with patients with HD. The prevalence of congestive heart failure (*p* < 0.05) and malignancy (*p* < 0.05) was also lower in the group with PD. In other variables, no significant differences were observed between the groups.

### 3.2. Biochemical Data

Biochemical data are shown in Table 2. Patients who selected PD had higher serum levels of albumin (3.3 vs. 3.2 g/dL, *p* < 0.05), GNRI (96 vs. 91, *p* < 0.01), and potassium (4.9 vs. 4.6 mEq/L, *p* < 0.05), while serum urea nitrogen level was lower in the group with PD (75.7 vs. 87.2 mg/dL, *p* < 0.01). There were no significant differences in serum levels of creatinine (8.6 vs. 8.8 mg/dL, *p* = 0.81), corrected calcium (8.5 vs. 8.6 mg/dL, *p* = 0.17), phosphorus (6.2 vs. 6.5 mg/dL, *p* = 0.28), triglyceride (134 vs. 116 mg/dL, *p* = 0.14), HDL–cholesterol (39 vs. 42 mg/dL, *p* = 0.98) and LDL–cholesterol (90 vs. 85 mg/dL, *p* = 0.50), hemoglobin level (10.0 vs. 9.7 g/dL, *p* = 0.22), and eGFR (5.4 vs. 4.9 mL/min/1.73 m^2^, *p* = 0.14) between PD and HD groups. In addition, no significant difference in eGFR decline rate for 6 months before dialysis initiation was observed between the two groups (42.4 vs. 45.0%, *p* = 0.31).

### 3.3. Timing of Education

The associations of the timing of education and dialysis modality selection are shown in Table 3. Patients with PD had higher eGFR at RRT education (8.9 vs. 7.9 mL/min/1.73 m^2^, *p* < 0.01) and a longer period from RRT education to dialysis initiation (6 vs. 4 months, *p* < 0.05). Patients who received RRT education primarily from nurse specialists selected PD compared with those who received that from nephrologists (18.6 vs. 9.0%, *p* < 0.05) (Appendix A). On the other hand, there were no significant differences between patients with PD and HD in eGFR at general education on CKD and a period from general education on CKD to dialysis initiation (9.3 vs. 8.9 mL/min/1.73 m^2^, *p* = 0.13 and 7 vs. 6 months, *p* = 0.20, respectively). These results suggest the importance of acquisition of knowledge on RRT rather than the general one on CKD to accomplish non-biased dialysis modality selection.

### 3.4. Influence of the Timing of RRT Education on Dialysis Modality Selection

Univariate logistic regression analysis for PD selection is shown in Appendix A. High GNRI (OR, 1.41; *p* < 0.01), and high eGFR at RRT education (OR, 1.16; *p* < 0.01) were a significant predictor of PD selection, whereas old age (OR, 0.63; *p* < 0.01) and high CCI (OR, 0.72; *p* < 0.01) were significantly associated with a decrease in PD selection. Female sex (OR, 0.69; *p* = 0.29), welfare public assistance (OR, 0.47; *p* = 0.47), living alone (OR, 0.70; *p* = 0.39), high eGFR at first visit to the nephrology department (OR, 1.01; *p* = 0.49), rapid eGFR decline rate for 6 months before dialysis initiation (OR, 0.95; *p* = 0.46) and high eGFR at dialysis initiation (OR, 1.04; *p* = 0.60) were not significantly associated with PD selection. On multivariate analyses using model 1, high eGFR at RRT education (OR, 1.14; *p* < 0.05) resulted in a significant predictor of PD selection (Table 4). On the other hand, old age (OR, 0.72; *p* < 0.01) and high CCI (OR, 0.78; *p* < 0.05) score were significantly associated with a decrease in PD selection. The goodness of fit of this model was sufficiently high because the Hosmer-Lemeshow test had a *p*-value of 0.58. In model 2, high eGFR at RRT education (OR, 1.12; *p* < 0.05) continued to significantly predict PD selection, whereas rapid eGFR decline rate for 6 months before dialysis initiation (OR, 0.95; *p* = 0.60) did not. Model 3 also showed that there is a trend that eGFR at RRT education (OR, 1.12; *p* = 0.07) was associated with PD selection; eGFR at dialysis initiation had no association (OR, 1.04; *p* = 0.72). The Hosmer-Lemeshow test had *p*-values of 0.84 in model 2 and 0.83 in model 3, respectively. There was no multicollinearity because variance inflation factor for all predictor variables was <5.0.

### 3.5. Dialysis Modality Selection and Timing of Patient Education for Each Attending Doctor

There were 15 attending doctors, all of whom had more than 10 years of experience as nephrologists. In order of the number of cases in charge, attending doctors were assigned from Dr. A to Dr. O, respectively. The differences in dialysis modality selection between attending doctors are shown in Appendix A. The PD selection rates varied from 0.0% to 54.5%, with a median of 10.0%. The timing of patient education for attending doctors is shown in Appendix A. There appeared to be a lot of variation in eGFR at RRT education and general education on CKD, not only among doctors but also among patients treated by the same doctor.

## 4. Discussion

To our best knowledge, this is the first clinical study to focus on the effect of the timing of RRT education on dialysis modality selection. The PD selection rate in this study was 14.9%, which was is lower than the 29–38% considered appropriate by most nephrologists, but comparable to the 8–48% observed in previous studies [11,12,20,21,22,23,24,25,26]. Our multivariate results using model 1 and model 2 showed that a higher eGFR at RRT education was significantly associated with a high rate of PD selection, as well as young age, and low CCI. In model 3, eGFR at RRT education tended to be associated with an increase in PD selection, though the association was not significant, probably due to a lack of power. On the other hand, there were no associations in eGFR decline rate for 6 months before dialysis initiation and eGFR at dialysis initiation with PD selection. Welfare public assistance also had no association with dialysis modality selection. This is possibly influenced by the fact that in Japan’s statutory health insurance system, providing universal coverage, out-of-pocket spending is extremely low regardless of the type of dialysis therapy or income level.

While many previous studies have reported that pre-dialysis RRT education increases the chances of PD preference, the effect of RRT education timing on PD selection has never been investigated [22,23,24,25,26]. In the present study, we observed that high eGFR at RRT education was associated with non-biased dialysis modality selection. We suggest three possible hypotheses for this finding. First, as renal function declines, it could become difficult for patients to have a comprehensive understanding of the differences of dialysis modalities and the effect of dialysis on their daily lives. In particular, cognitive impairment may be associated with biased dialysis modality selection. Although the underlying pathophysiological mechanisms of cognitive impairment in CKD have not been fully understood, cardiovascular and nephrogenic factors have been proposed as candidate etiologies [27,28]. Cognitive impairment occurs in the early CKD stage, whereas its frequency and severity increase according to the decline in renal function [27,29]. CKD associated with cognitive impairment has the characteristic of a deficit in executive functions accompanied by impaired decision-making capacity [29]. Patients who received RRT education later could not fully appreciate RRT education, and thus could make inappropriate decisions on dialysis modality selection. Indeed, Zee, et al. reported that patients selecting PD as RRT tended to be more engaged in the dialysis modality decision-making process and were satisfied with the modality they chose [30]. Since PD is often self-managed, high acceptance of disease and increased self-efficacy is important. In addition, uremic symptoms, such as heart failure associated with the progression of renal dysfunction, could also lead to impaired decision-making capacity and make it hard for patients to select RRT in an appropriate manner [31]. Another possibility is that the low percentage of PD selection might be due to an unexpected dialysis start. Delays in RRT education result in an increase in unexpected dialysis initiation and, in this case, HD tends to be favored in many dialysis centers than PD [32,33]. By reducing the likelihood of unscheduled dialysis beginning, early RRT education might lead to non-biased dialysis modality selection. Finally, it can be due to the attending doctors’ preconceived notions of which dialysis modality is appropriate for the patient. Because of the significant effect of residual renal function on prognosis in patients undergoing PD, patients who select PD are recommended to initiate dialysis earlier compared with those who select HD: Attending nephrologists may have implemented RRT education by nurse specialists with time to spare for patients who they subjectively judged to be suitable for PD [34]. The large variation in the timing of RRT education with the same doctors, and the low rates of PD selection when RRT education was provided by nephrologists themselves in this study, might reflect this preconception.

In the present study, young age, and low CCI were also significant predictors of high rates of PD selection, which is consistent with previous reports [11,12,13,21]. Jager et al. discovered a stronger preference for HD in patients with poor activities of daily living or physical weakness [22]. Medical personnel, on the other hand, could be responsible for the low rate of PD selection. Naturally, they should provide all patients with equal information concerning RRT, but it is not uncommon for them to deliver biased RRT education that favors HD over PD, especially for patients with low activity and daily life handicaps due to a variety of causes [35,36]. Some of the health care providers at our facility, although hard to verify, may have performed this type of inappropriate education. Actually, the PD selection rate by attending doctors in this study varied greatly. In this situation, assisted PD by trained staff or family members has recently attracted a lot of attention. Several studies demonstrated that assisted PD presented equality or superiority in mortality and peritonitis risk compared with self-care PD [37,38]. Furthermore, assisted PD can be beneficial in risk of transfer to HD, especially to frail patients with low exercise capacity [37,39]. The Frail and Elderly Patient Outcomes on Dialysis study demonstrated that the quality of life of elderly patients on assisted PD was similar to those of patients receiving in-center HD, and treatment satisfaction was higher on assisted PD [40]. Additionally, PD, which is characterized by continuous removal of extracellular fluid and the absence of arteriovenous access, has advantages in terms of circulatory dynamics [3,4]. PD also has the potential to preserve residual renal function, leading to a better patient prognosis of mortality compared with HD [5,41,42]. These findings show that, in the case of assisted PD, advanced age or the presence of several comorbidities, may be even more favorable considerations in PD selection, and that medical personnel should give unbiased RRT education.

This study has some major limitations that should be mentioned. First, because this was a retrospective observational study, these results should be interpreted cautiously and the cause-and-effect relationship between RRT timing and PD selection is unclear. The results of this study might merely reflect the recommendation that PD patients should initiate maintenance dialysis at higher eGFR compared with HD patients, although the eGFR at dialysis initiation was adjusted in the multivariate analysis. Although there were statistically significant differences between patients with PD and HD in age and CCI, the multivariate models also included both of them as independent variables. Data on various confounding factors such as education level or regular employment were unavailable, and accordingly not included in the multivariate models. Additionally, the difference in the way medical staff gave RRT education could not be considered. Considering the differences in PD selection rate between attending doctors, some of them may have given biased RRT education based on their preconception. Because we gathered data by reviewing computerized medical records, the likelihood of information bias could not be ruled out. As a countermeasure, two authors independently checked the data to minimize this bias. Second, despite our hospital’s vast medical region, selection bias may have occurred in this single-center design with only 355 participants. RRT education seems to be delayed compared to the average. Patients who chose PD received renal replacement therapy education at eGFR only 1 mL/min/1.73 m^2^ higher (2 months earlier) than those who chose hemodialysis, although the difference was significant. Uremic symptoms in patients with ESRD can rapidly worsen over a short period of time; therefore, only a small difference in eGFR and timing on RRT education may have produced a statistically significant difference in RRT selection under the condition of our overall late start of RRT education. To elucidate the causality and strengthen the generalizability, multicenter prospective studies are needed. Despite the aforementioned limitations, this study is unique in that it evaluated the effect of RRT education timing on dialysis modality selection, supporting the importance of early RRT education.

## 5. Conclusions

Not only young age and low CCI, but also high eGFR at RRT education, was significantly associated with the increased chance for PD selection. To improve biased dialysis modality selection, it may be desirable to provide RRT education early to the extent possible.

## Figures and Tables

**Figure 1 jcm-11-04042-f001:**
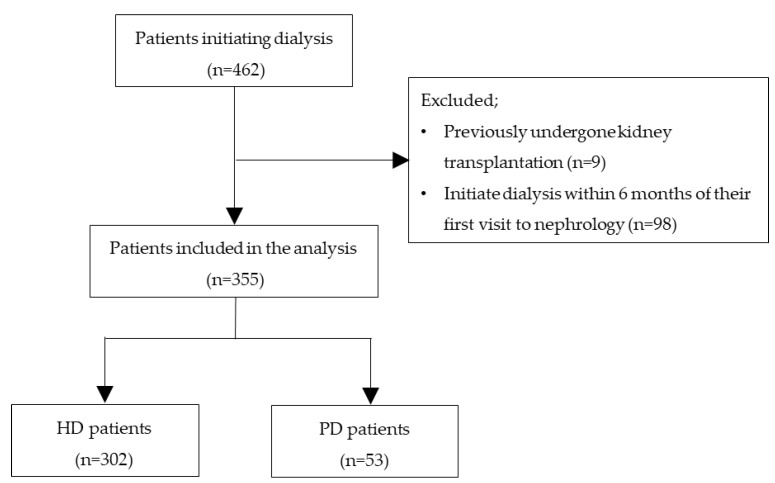
Flow chart of the present study.

**Table 1 jcm-11-04042-t001:** Patients’ baseline characteristics in groups with HD and PD.

Variables	All Patients (*n* = 355)	Group with HD (*n* = 302)	Group with PD (*n* = 53)	*p*-Value
Age (year)	70 (59–79)	71 (61–80)	59 (50–71)	<0.01
Sex (% female)	102 (28.7%)	90 (29.8%)	12 (22.6%)	0.37
Welfare public assistance recipient (%)	13 (3.7%)	12 (4.0%)	1 (1.9%)	0.46
Living alone (%)	69 (19.4%)	61 (20.2%)	8 (15.1%)	0.39
Smoking history	207 (58.3%)	178 (58.9%)	29 (54.7%)	0.67
Body mass index	23.2 (20.4–26.1)	23.1 (20.2–25.7)	23.9 (22.2–27.2)	<0.05
Systolic blood pressure (mmHg)	139.2 ± 19.6	139.7 ± 19.8	136.3 ± 18.3	0.26
Diastolic blood pressure (mmHg)	74.9 ± 14.9	74.3 ± 15.2	77.8 ± 12.5	0.12
Charlson comorbidity index	4 (3–6)	5 (4–6)	4 (3–4)	<0.01
Underlying conditions				
Diabetic kidney disease	130 (36.6%)	108 (35.8%)	22 (41.5%)	0.52
Renal sclerosis	88 (24.8%)	80 (26.5%)	8 (15.1%)	0.11
Glomerulonephritis	59 (16.6%)	44 (14.6%)	15 (28.3%)	<0.05
Polycystic kidney disease	14 (3.9%)	11 (3.6%)	3 (5.7%)	0.75
Tubulointerstitial nephritis	15 (4.2%)	14 (4.6%)	1 (1.9%)	0.58
Others	49 (13.8%)	45 (14.9%)	4 (7.5%)	0.22
Comorbidities				
Diabetes	160 (45.1%)	131 (43.4%)	29 (54.7%)	0.17
Hypertension	334 (94.1%)	282 (93.4%)	52 (98.1%)	0.30
Coronary artery disease	63 (17.7%)	54 (17.9%)	9 (17.0%)	1.00
Congestive heart failure	98 (27.6%)	90 (29.8%)	8 (15.1%)	<0.05
Cerebrovascular disease	65 (18.3%)	59 (19.5%)	6 (11.3%)	0.22
Malignancy	82 (23.1%)	76 (25.2%)	6 (11.3%)	<0.05
eGFR at first visit to the nephrology department (mL/min/1.73 m^2^)	25.2 (16.8–37.6)	24.9 (16.9–37.1)	25.6 (14.1–42.1)	0.89

Abbreviations: HD, hemodialysis; PD, peritoneal dialysis; eGFR, estimated glomerular filtration rate.

**Table 2 jcm-11-04042-t002:** Biochemical data of groups with patients with HD and PD at dialysis initiation.

Variables	All Patients (*n* = 355)	Group with HD (*n* = 302)	Group with PD (*n* = 53)	*p*-Value
Urea nitrogen (mg/dL)	85.3 (70.0–102.5)	87.2 (71.6–104.4)	75.7 (62.5–88.8)	<0.01
Creatinine (mg/dL)	8.8 (7.4–10.5)	8.8 (7.4–10.5)	8.6 (7.5–9.9)	0.81
eGFR (mL/min/1.73 m^2^)	5.0 (4.0–6.1)	4.9 (3.9–6.1)	5.4 (4.4–6.1)	0.14
Albumin (g/dL)	3.2 ± 0.6	3.2 ± 0.6	3.3 ± 0.5	<0.05
Geriatric nutritional risk index	92 (83–100)	91 (82–99)	96 (90–103)	<0.01
Hemoglobin (g/dL)	9.8 ± 1.4	9.7 ± 1.4	10.0 ± 1.2	0.22
Potassium (mEq/L)	4.6 ± 0.9	4.6 ± 0.9	4.9 ± 0.8	<0.05
Corrected calcium (mg/dL)	8.6 ± 1.1	8.6 ± 1.2	8.5 ± 0.5	0.17
Phosphorus (mg/dL)	6.5 ± 1.9	6.5 ± 2.0	6.2 ± 1.4	0.28
Triglyceride (mg/dL)	118 (84–156)	116 (83–155)	134 (88–175)	0.14
HDL–cholesterol (mg/dL)	42 (33–54)	42 (33–53)	39 (33–53)	0.98
LDL–cholesterol (mg/dL)	87 (67–110)	85 (67–105)	90 (74–109)	0.50
eGFR decline rate for 6 months before dialysis initiation (%)	44.5 (31.2–57.3)	45.0 (31.3–57.6)	42.4 (31.0–53.9)	0.31

Abbreviations: HD, hemodialysis; PD, peritoneal dialysis; eGFR, estimated glomerular filtration rate; HDL, high–density lipoprotein; LDL, low–density lipoprotein.

**Table 3 jcm-11-04042-t003:** Timing of education and choice of dialysis modality.

Variables	All Patients (*n* = 355)	Group with HD (*n* = 302)	Group with PD (*n* = 53)	*p*-Value
eGFR at RRT education (mL/min/1.73 m^2^)	8.0 (6.3–9.7)	7.9 (6.2–9.5)	8.9 (7.4–11.0)	<0.01
eGFR at general education on CKD (mL/min/1.73 m^2^)	9.1 (6.9–11.7)	8.9 (6.5–11.5)	9.3 (8.0–11.8)	0.13
Time from RRT education to dialysis initiation (month)	4 (2–9)	4 (2–9)	6 (4–10)	<0.05
Time from general education on CKD to dialysis initiation (month)	6 (2–16)	6 (1–17)	7 (4–11)	0.20

Abbreviations: HD, hemodialysis; PD, peritoneal dialysis; eGFR, estimated glomerular filtration rate; RRT, renal replacement therapy; CKD, chronic kidney disease.

**Table 4 jcm-11-04042-t004:** Results of multivariate logistic regression analyses associated with PD selection.

Variables	Model 1		Model 2		Model 3	
	OR (95% CI)	*p*-Value	OR (95% CI)	*p*-Value	OR (95% CI)	*p*-Value
Age (per 10 years)	0.72 (0.57–0.91)	<0.01	0.70 (0.55–0.90)	<0.01	0.70 (0.55–0.90)	<0.01
Sex (female)	0.74 (0.34–1.60)	0.44	0.72 (0.33–1.56)	0.40	0.74 (0.33–1.63)	0.45
Welfare public assistance	0.73 (0.08–6.33)	0.78	0.70 (0.08–6.12)	0.75	0.70 (0.08–6.09)	0.74
Living alone	0.58 (0.23–1.46)	0.25	0.60 (0.24–1.50)	0.27	0.60 (0.24–1.51)	0.28
Charlson comorbidity index (per 1)	0.78 (0.63–0.96)	<0.05	0.78 (0.63–0.96)	<0.05	0.77 (0.62–0.96)	<0.05
Geriatric nutritional risk index (per 10)	1.25 (0.96–1.62)	0.10	1.22 (0.93–1.60)	0.15	1.24 (0.94–1.64)	0.14
eGFR at first visit to the nephrology department (per 1 mL/min/1.73 m^2^)	1.00 (0.99–1.02)	0.83	1.00 (0.99–1.02)	0.84	1.00 (0.98–1.02)	0.88
eGFR at RRT education (per 1 mL/min/1.73 m^2^)	1.14 (1.02–1.27)	<0.05	1.12 (1.00–1.26)	<0.05	1.12 (0.99–1.26)	0.07
eGFR decline rate for 6 months before dialysis initiation (%)	–	–	0.95 (0.80–1.14)	0.60	0.97 (0.79–1.20)	0.81
eGFR at dialysis initiation (per 1 mL/min/1.73 m^2^)	–	–	–	–	1.04 (0.84–1.29)	0.72

Abbreviations: PD, peritoneal dialysis; OR, odds ratio; CI, confidence interval; eGFR, estimated glomerular filtration rate; RRT, renal replacement therapy.

## Data Availability

The data presented in this study are available on request from the corresponding author.

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
