# Peer review of "Late Dialysis Modality Education Could Negatively Predict Peritoneal Dialysis Selection"

_jcm, 2022, doi:10.3390/jcm11144042_

Round 1

Reviewer 1 Report

Nakayama and collaborators in this retrospective study highlight the importance of renal replacement therapy education in determining the type of dialysis chosen by the patient. This retrospective observational study included about 355 patients of whom only 53 patients chose peritoneal dialysis as a dialysis technique. Data analysis showed that the choice of peritoneal dialysis was favored if dialysis education was done early and the patients had  a higher glomerular flitrate. Although the study is clear and well written it presents some biases.  The most important problem in this study is that in the clinical practice  peritoneal dialysis is initiated earlier than  extracorporeal dialysis and the filtrate is generally higher than in patients who are initiated on extracorporeal dialysis. It is therefore obvious that most of the patients who choose PD were educated early and had a higher glomerular filtratio rate  at the start of educational visits. To answer the main question of the study, the authors should have carried out  a prospective study and only in this way they could have obtained reliable results.

Moreover, the authors do not report anything about  the interview they had with the patients and therefore it is not possible to assess whether the patients were in any way influenced by the doctors. It’s also obvious that the population that choose PD was younger because younger subjects  are usually more autonomous and have fewer clinical problems and fewer comorbidities. It seems to me that the results of the study are obvious and that the study does not add anything new to what we all know about choosing the type of dialysis.

Author Response

Response to Reviewer 1:

First and foremost, we appreciate your valuable comments. To address these concerns, we have revised the original manuscript accordingly. The revised sentences are indicated as blue text in the revised manuscript.

1) Nakayama and collaborators in this retrospective study highlight the importance of renal replacement therapy education in determining the type of dialysis chosen by the patient. This retrospective observational study included about 355 patients of whom only 53 patients chose peritoneal dialysis as a dialysis technique. Data analysis showed that the choice of peritoneal dialysis was favored if dialysis education was done early and the patients had a higher glomerular filtrate. Although the study is clear and well written it presents some biases. The most important problem in this study is that in the clinical practice peritoneal dialysis is initiated earlier than extracorporeal dialysis and the filtrate is generally higher than in patients who are initiated on extracorporeal dialysis. It is therefore obvious that most of the patients who choose PD were educated early and had a higher glomerular filtration rate at the start of educational visits. To answer the main question of the study, the authors should have carried out a prospective study and only in this way they could have obtained reliable results.

Response: As you pointed out, patients who select peritoneal dialysis (PD) are generally recommended to start maintenance dialysis earlier than those who select hemodialysis (HD). Therefore, attending doctors may have implemented renal replacement therapy (RRT) education early for patients who they subjectively judged to be suitable for PD, leading to the results obtained in this study. A prospective interventional study is essential to elucidate the cause-and-effect relationship between RRT timing and dialysis modality selection. We have placed additional emphasis on this very important limitation in the Discussion (lines 258–263, 288–293, 306 and 307). On the contrary, in this study, no significant difference was found in eGFR at dialysis initiation between the PD group and the HD group (Table 2). Furthermore, the multivariate analysis including eGFR at dialysis initiation as a covariate (Table 4) demonstrated that early RRT education tended to be associated with an increase in PD selection. We believe that early RRT education itself could be beneficial in non-biased dialysis modality selection.

2) Moreover, the authors do not report anything about the interview they had with the patients and therefore it is not possible to assess whether the patients were in any way influenced by the doctors.

Response: How doctors (and/or nurses) describe PD and HD to the patients is an important point. Although this is a single-center study, the specific description of dialysis modality was likely to be influenced by the preference of medical staff. Essentially, in this study, the PD selection rate by attending doctors highly varied (Figure S1) (lines 207–209). However, given the nature of the observational study, we could not evaluate this in detail. We have revised relevant parts of the Discussion to emphasize this point (lines 294–296).

3) It’s also obvious that the population that choose PD was younger because younger subjects are usually more autonomous and have fewer clinical problems and fewer comorbidities. It seems to me that the results of the study are obvious and that the study does not add anything new to what we all know about choosing the type of dialysis.

Response: This study aimed to evaluate the effect of RRT timing on dialysis modality selection. As you pointed out, a young age and a low Charlson comorbidity index have been reported as significant predictors of high rates of PD selection. Therefore, these were included into the multivariate models as confounders. We firstly demonstrated that the timing of RRT education could predict dialysis modality selection, although some observational studies have reported that RRT education is associated with the increased chance of PD selection. Certainly, new findings of this study might be not very remarkable, but we believe that they are of high clinical significance with novelty in supporting the importance of early RRT education. The publicaion of this study in the Journal of Clinical Medicine could motivate medical staff to provide RRT education at an early stage.

Reviewer 2 Report

1. Please clarify about education that who provided the education? Physicians or nurses or educators? Any check list that you had, please attach in supplementary

2. How did the investigators ascertain that education has been provided.

3. Patient's education level (college degree or high school etc) is also confounding factors

Author Response

Response to Reviewer 2:

We would like to extend our sincere gratitude for providing your valuable comments. To address your concerns, we have performed additional analyses and have revised the original manuscript. The revised sentences are indicated as blue text in the revised version.

1) Please clarify about education that who provided the education? Physicians or nurses or educators? Any check list that you had, please attach in supplementary.

Response: As you pointed out, it is a useful information who provided RRT education. Therefore, we have created new Table S1, and revised the Materials and Methods, Results and Discussion (lines 86, 87, 164–166, 258–263, 317 and 318).

2) How did the investigators ascertain that education has been provided.

Response: Thank you for your valuable comment. Our original manuscript was difficult to understand, especially on how we ascertain RRT education was provided to the patients. We have revised the Materials and Methods to clarify this (lines 83, 84).

3) Patient’s education level (college degree or high school etc) is also confounding factors

Response: We completely agree with your comments. Patients’ education level is an important confounding factor. Regrettably, we could not obtain data on patients’ education level from the electronic medical records, which is one of the limitations of this study. Instead, we have collected the information on whether patients received welfare public assistance or lived alone as socioeconomic factors and included these in the multivariate models. Accordingly, we have revised Table 1, Table 4, the Materials and Methods, Results and Discussion to describe the above (lines 24–29, 68–73, 107–109, 125, 126, 182–197, 224–228, 293 and 294).

Reviewer 3 Report

The authors explore the feasibility of choosing PD over HD with respect to the timing of dialysis education. This is both an interesting and novel point of view. They study many variables to achieve their goal.

Although it is a retrospective study, the size of the study cohort is quite large to draw conclusions.  The text is well written and discussed.

My comments to the work are:

- The low serum calcium level in both groups is striking. I suggest that their baseline levels appear in the table.

- The authors state that patients on PD had only 1 ml/min higher eGFR at the time of RRT education than on HD and a longer period (only two months) from RRT education to the start of dialysis. Although statistically significant, I cannot understand why these small differences between the two groups might influence the choice of one dialysis modality over the other.

- The eGFRs of both groups are quite low, so in my opinion it is excessive to speak of "early education". In fact, I believe that an eGFR of 7-8 ml/min should be considered late education.

- What did the education consist of?

- The authors say that education level, income level, regular employment or roommates were excluded from the analysis. In my opinion it would be interesting to know the relationship of these variables on the choice of RRT modality.

Finally, I am grateful for the opportunity to review this interesting work.

Author Response

Response to Reviewer 3:

First and foremost, we appreciate your valuable comments. To address these concerns, we have performed additional analyses and have revised the original manuscript accordingly. The revised sentences are indicated as blue text in the revised manuscript.

1) The low serum calcium level in both groups is striking. I suggest that their baseline levels appear in the table.

Response: In our original manuscript, serum calcium levels were not adjusted for hypoalbuminemia, leading to misunderstanding. Therefore, we have revised Table 2, the Materials and Methods and Results to describe corrected calcium levels using Payne’s formula (lines 78–79, 146–151, 367 and 368).

2) The authors state that patients on PD had only 1 ml/min higher eGFR at the time of RRT education than on HD and a longer period (only two months) from RRT education to the start of dialysis. Although statistically significant, I cannot understand why these small differences between the two groups might influence the choice of one dialysis modality over the other.

Response: Thank you for your valuable comments. In patients with end-stage renal disease, uremic symptoms including heart failure, decreased motivation, loss of appetite, or cognitive impairment can dramatically exacerbate than expected. As pointed out in your third comment, the timing of RRT education in this study appears to be relatively late, and patients’ conditions could have easily changed in just a few months. Therefore, these slight differences in eGFR and timing on RRT education have led to significant difference in RRT selection under the condition of our late start in RRT education. We have discussed this point in the Discussion (lines 300–306).

3) The eGFRs of both groups are quite low, so in my opinion it is excessive to speak of "early education". In fact, I believe that an eGFR of 7-8 ml/min should be considered late education.

Response: We completely agree with your comments. The sentence “early RRT education” appears to be inappropriate to precisely describe the context of the present study. We have clarified this point throughout the paper (lines 2, 3, 21, 22, 29, 30, 187, 188, 231, 232, 258–260 and 311–314).

4) What did the education consist of?

Response: Thank you for your question. RRT education was provided by either or both nephrologists and/or nurse specialists and consisted of different forms of dialysis therapy and their detailed features including the effect on daily life or possible complications (lines 87–89). On the contrary, general education on chronic kidney disease (CKD) (except RRT education) provided by nurse specialists consisted of dietary habits, medication adherence, living environment modifications, and future uremia symptoms (lines 89–94).

5) The authors say that education level, income level, regular employment or roommates were excluded from the analysis. In my opinion it would be interesting to know the relationship of these variables on the choice of RRT modality.

Response: We wholeheartedly agree with you. We ascertained whether patients were living alone and included this into the multivariate models. Data on education level or regular employment were not unavailable from our electronic medical records. Detailed income level was not obtained either; however, we collected information about whether patients received welfare public assistance instead (i.e., those whose household income is below the minimum cost of living). Accordingly, we added this as a covariate in the multivariate models. We have revised Table 1, Table 4, the Materials and Methods, Results, and Discussion to specify the above (lines 24–29, 68–73, 107–109, 125, 126, 182–197, 224–228, 293 and 294).

Other modifications

1) We have revised the row for creatinine in Table 2 to make all decimal points equal.

2) Based on the instruction from English proofreading through the proofreading company Enago (https://www.enago.jp), we have replaced the term “impact” with “effect” to use more precise terms (lines 47, 48, 51, 52, 104–106, 215, 216, 229–231, 233–235, 256–258, and 307–309).

3) We deeply apologize that “eGFR at first visit to the nephrology department” was not included in original Table 1. We have added this item in Table 1.